# Neuroendocrine and Inflammatory Effects of Childhood Trauma Following Psychosocial and Inflammatory Stress in Women with Remitted Major Depressive Disorder

**DOI:** 10.3390/brainsci9120375

**Published:** 2019-12-13

**Authors:** Laura L.M. Cassiers, Peter Niemegeers, Erik Fransen, Manuel Morrens, Peter De Boer, Luc Van Nueten, Stephan Claes, Bernard G.C. Sabbe, Filip Van Den Eede

**Affiliations:** 1Collaborative Antwerp Psychiatric Research Institute, University of Antwerp, Wilrijk 2610, Belgium; peter.niemegeers@student.uantwerpen.be (P.N.); manuel.morrens@uantwerpen.be (M.M.); Filip.VanDenEede@uza.be (F.V.D.E.); 2University Department of Psychiatry, Campus Antwerp University Hospital, Edegem 2650, Belgium; 3Department of Psychiatry, ZNA Hospitals, Antwerp, Antwerp 2060, Belgium; 4StatUa Center for Statistics, University of Antwerp, Wilrijk 2610, Belgium; erik.fransen@uantwerpen.be; 5University Department of Psychiatry, Campus Psychiatric Hospital Duffel, Duffel 2570, Belgium; 6Janssen Research & Development, Janssen Pharmaceutica N.V., Beerse 2340, Belgium; PDEBOER1@its.jnj.com (P.D.B.); LVNUETEN@its.jnj.com (L.V.N.); 7University Psychiatric Centre KU Leuven, Kortenberg 3070, Belgium; stephan.claes@upckuleuven.be

**Keywords:** “Child Abuse” [Mesh], “Depressive Disorder” [Mesh], “Pituitary-Adrenal System” [Mesh], inflammation, psychosocial stress, vulnerability

## Abstract

The dysregulation of the inflammatory and neuroendocrine systems seen in major depressive disorder (MDD) may persist after remission and this is associated with a higher risk of relapse. This vulnerable subgroup may be characterized by a history of childhood trauma. In a single-blind randomized placebo-controlled crossover study, 21 women with remitted recurrent MDD and 18 healthy controls were exposed to psychosocial stress (Trier social stress test) or inflammatory stress (typhoid vaccine), or both, to investigate the effects of childhood trauma on the neuroendocrine and inflammatory responses. Childhood trauma was assessed using the Childhood Trauma Questionnaire and participants were dichotomized into a traumatized and non-traumatized group. Serum adrenocorticotropic hormone (ACTH), cortisol, interferon (IFN)-γ, tumor necrosis factor (TNF)-α, and interleukin (IL)-6 were measured at regular intervals after each intervention. The effects of trauma, time, and intervention on these parameters were modeled by fitting linear mixed models. Childhood trauma in itself did not have a main effect on the outcome measurements. However, an interactional effect of trauma with stressor type was found in the remitted MDD group: trauma was associated with higher cortisol levels only after adding immunological to psychosocial stress, and with lower TNF-α levels in response to vaccination. This suggests the existence of a vulnerable trauma-associated MDD endophenotype.

## 1. Introduction

Major depressive disorder (MDD) is a leading cause of disease burden worldwide [1]. Stressful life events, particularly those involving social rejection, are associated with a higher risk for MDD, possibly through their triggering of stress and immune response systems that are also dysregulated in MDD [2,3]. Indeed, MDD is characterized by a chronic hyperactivity of the hypothalamic–pituitary–adrenal (HPA) axis, an important component of the stress system [4,5,6]. Elevated cortisol levels are reported in MDD, as well as a reduced suppression after the administration of synthetic glucocorticoids (i.e., a dexamethasone suppression test (DST)) [2], and an enhanced response to the administration of corticotropin-releasing factor (CRF) after dexamethasone suppression (DEX/CRF) [7,8] and to psychosocial stress [9]. Major depressive disorder is also associated with increased plasma levels of proinflammatory cytokines, such as interleukin (IL)-6 and tumor necrosis factor (TNF)-α, that may play a role in the disease’s pathogenesis [2].

These disturbances of the stress and immune systems may or may not persist after remission of the major depressive episode. Some studies report HPA axis hyper(re)activity in remitted MDD patients [10,11,12], whereas others failed to demonstrate significant differences compared to healthy controls [13,14,15,16,17]. As to the immune system, evidence exists of a persistent proinflammatory state in remitted MDD patients [18,19].

In a study by our group with the combined DEX/CRF test, the HPA axis function remained disturbed only in a subgroup of patients with remitted MDD [20], suggesting the existence of a neurobiologically distinct subgroup. Notably, patients with remitted MDD and persisting HPA axis disturbances are at higher risk for a recurrent major depressive episode [21,22,23,24]. Childhood trauma, a well-known vulnerability factor for the development of mood disorders [25,26], apparently confounds this association [22]. Not surprisingly, childhood trauma in itself is associated with a dysregulation of the stress system. An initial hyperactivation of the HPA axis following adverse childhood experiences appears to be adaptively downregulated in the long term, leading to lower activity [27,28,29,30].

Taken together, these findings seem to suggest that previously depressed individuals who have persistent HPA axis abnormalities (and therefore are at higher risk for recurrence) might be part of a vulnerable patient subgroup that also experienced trauma in childhood. Heim and colleagues [31] previously demonstrated the existence of a biologically distinct trauma endophenotype within a population of currently depressed patients. In this study, HPA axis hyperreactivity to a DEX/CRF test was only present in depressed subjects reporting childhood abuse, and not in those without such a history.

The trauma endophenotype also distinguishes itself at the immunological level, as the association between a proinflammatory state and depression was shown to disappear after controlling for childhood maltreatment [32]. Indeed, a history of childhood trauma is in itself associated with a proinflammatory state [33,34], with elevations of cytokines IL-6 and TNF-α [35,36]. To date however, little is known about the stress system and immunological manifestation of this endophenotype in remitted MDD.

In a previous paper by our research group, HPA axis and inflammatory responses to a psychosocial (Trier social stress test, TSST) and immunological stressor (typhoid vaccine) were described in (partially) remitted recurrent MDD patients [37]. The psychosocial stressor was found to activate the HPA axis in both MDD patients and healthy controls as well as lead to decreased inflammatory cytokines, which was suggested to result from cortisol’s anti-inflammatory effect. The typhoid vaccine was associated with a trend for a general rise of adrenocorticotropic hormone in control individuals, but not in the MDD patients. The combined psychosocial and immunological stressor led to an increase of proinflammatory cytokine interferon (IFN)-γ, but only in the remitted MDD group.

The objective of the current analyses is to further assess the effect of childhood trauma on the stress and immune responses to a psychosocial and immunological stressor in this well-described sample of remitted recurrent MDD patients. We hypothesize that childhood trauma will be associated with differential cortisol and adrenocorticotropic hormone (ACTH) responses to a psychosocial stressor in remitted MDD patients. Given the proinflammatory state associated with childhood trauma and the fact that immune cells can become resistant to the anti-inflammatory effects of glucocorticoids when facing acute psychosocial stress [3], we also expect levels of inflammatory cytokines to be higher after both stressors.

## 2. Materials and Methods

### 2.1. Subjects

A total of 21 patients with (partially) remitted recurrent major depressive disorder (MDD) participated in the study, as well as 18 controls, aged between 25 and 45 years. Patients were recruited from the participating psychiatric hospitals, as well as from neighboring primary and secondary (mental) healthcare centers. Never-depressed controls were recruited through advertisement. All participants were female in order to rule out gender-based differences in stress and inflammatory responses [38,39,40]. All participants provided informed consent. Patients had moderate-to-severe recurrent MDD without psychotic features according to the Diagnostic and Statistical Manual of Mental Disorders, fourth edition, text revised (DSM-IV-TR) [41] criteria. Their most recent depressive episode was within the last 24 months, but with a minimum three-month stable period of at least partial remission (Montgomery–Åsberg depression rating scale [42] score < 15). During the study, a total of three patients were lost at follow-up and replaced.

Patients were required to have stable vital parameters, clinical examination, and blood and urine tests, as well as a body mass index between 18 and 30 kg/m^2^. Other DSM-IV Axis I diagnoses, as well as recent (≤6 months) substance use disorders (with the exception of nicotine and caffeine), acute suicidality, or a history of serious illness were reasons for exclusion.

To maximize the comparability of stress and immunological responses, participants were not included if they had been previously exposed to the stress test used in this study or if they had recently (≤6 months) experienced severe psychosocial stress. Also, participants who received a typhoid vaccination within the previous five years were excluded. Pharmacological exclusion criteria were taking more than one antidepressant or any pharmacological agent affecting the immune system.

The study was conducted at the Antwerp University Department of Psychiatry, Campus Psychiatric Hospital Duffel, and the University Psychiatric Center KU Leuven, Campus Leuven, Belgium. Both the central and local Ethics Committees and the Belgian health authorities approved of the study protocol, which was in accordance with the regulations of the participating institutions and the International Conference on Harmonization—Good Clinical Practice Guidelines as well as the European Directive 2001/20/EC. The study was registered in the ClinicalTrials.gov database under the identifier (NCT01533285) (https://clinicaltrials.gov/ct2/show/NCT01533285) and the and European Clinical Trials (EudraCT) databases under the identifier 2011-004898-80.

### 2.2. Study Design

The current study has a randomized, single-blind, placebo-controlled, and crossover design. Subjects considered eligible for participation, both healthy control participants and participants with remitted MDD, were computer-randomized to one of six possible intervention sequences administered over the course of two study visits, separated by a seven to fourteen day washout period. The intervention sequences consisted of three possible active interventions (an isolated immunological or psychosocial stressor or a combined challenge) on the first visit, followed by placebo on the second visit or vice versa (Figure 1). Only participants were blinded as to which of the interventions they received.

At each visit, participants were first screened with an alcohol breath and urine drug and pregnancy test. The psychosocial challenge consisted of the Trier social stress test (TSST), in which the participant partakes in a five-minute fictitious job interview with only three minutes of preparation, followed by a five-minute arithmetic exercise in front of a critical jury. All the while, the participant must speak into a microphone and be video recorded [43]. The immunological stressor consisted of an intramuscular injection with the typhoid vaccine (0.5 mL containing 25 μg *Salmonella typhi* capsular polysaccharide; Typhim ^®®^ Vi; Sanofi Pasteur MSD, Diegem, Belgium). The placebo condition consisted of an intramuscular injection with 0.5 mL 0.9% NaCl. Depending on the allocated intervention sequence, the TSST was administered if applicable at 12:00 PM on the visit day, and at 12:20 PM, was followed by either the placebo injection or the typhoid vaccine from indistinguishable syringes. Regular monitoring of vital parameters happened throughout the testing periods and all measurements (up to the last one post-dose) were performed in the participating study centers.

### 2.3. Assessments

#### 2.3.1. Baseline Mood

Baseline mood was assessed using the MADRS [42] during an eligibility screening preceding the first study visit. The MADRS is a 10-item interview used to measure the severity of a depressive episode. Each item is scored on a scale from 0 to 6.

#### 2.3.2. Childhood Trauma

Adverse events in childhood were assessed in retrospect using the 28-item self-report Childhood Trauma Questionnaire (CTQ) [44]. In this questionnaire, participants report the occurrence and frequency of childhood abuse and neglect on a 6-point Likert scale, ranging from 0 (never true) to 5 (very often true). The questionnaire encompasses five abuse/neglect subscales (sexual abuse, physical abuse and neglect, and emotional abuse and neglect) of five items each, as well as a minimization/denial scale of three items.

#### 2.3.3. Biological Measures

Serum adrenocorticotropic hormone (ACTH) and cortisol levels were measured as markers of hypothalamic–pituitary–adrenal (HPA) axis reactivity. Likewise, levels of the inflammatory cytokines interleukin (IL)-6, interferon (IFN)-γ, and tumor necrosis factor (TNF)-α were determined as measures of the immunological response. On each visit day, subjects received a peripheral venous catheter from which blood samples were drawn at 11:55 AM (immediately before the TSST, if applicable) and 30, 60, 90, 150, 180, 240, and 360 min after the injection of either typhoid vaccine or placebo. Two additional blood samples were taken for ACTH and cortisol analyses immediately after the injection (at 12:21 PM) as well as 15 min post-intervention.

ACTH and cortisol levels were analyzed by immunoassay (Siemens^®^ IMMULITE 2000 Immunoassay System) at PRA International (Zuidlaren, the Netherlands), with detection ranges of 1.1–278 pmol/L for ACTH and 28–1380 nmol/L for cortisol. Inflammatory cytokine levels were determined by quantitative electrochemiluminescence immunoassays (Meso Scale Discovery^®^ V-PLEX Proinflammatory Panel 1 (human) kits) at Janssen Biobank (Beerse, Belgium) with detection ranges of 0.2–0.9 to 1060–1320 ng/L for IFN-γ, 0.06–0.3 to 320–352 ng/L for TNF-α and 0.07–0.3 to 743–833 ng/L for IL-6. Measurements outside the detection range were coded as missing.

#### 2.3.4. Statistical Analyses

Mean baseline group differences were assessed through the Student’s *t* test in case of normally distributed outcomes or using a Wilcoxon–Mann–Whitney test for non-normal outcomes. If values for the levels of the biological markers were below the detection range, the lowest quantifiable concentration was used. Pearson’s *χ*^2^ test was used to test the association between categorical variables. Analyses were run separately in the remitted MDD and the never-depressed control groups to assess the specificity of the hypothesized trauma effects to our remitted MDD group and to investigate potentially differential trauma effects in both groups.

To model the effect of trauma on the five biological measures (ACTH, cortisol, IL-6, IFN-γ, and TNF-α) in response to the neuroendocrine and inflammatory stressors in remitted MDD, we fitted linear mixed models. Given the non-normal distribution of the biological data, all were log-transformed. The assumptions of the linear mixed models (normality of the residuals and homoscedasticity) were checked by visual inspection of the residual quantile–quantile plots, and a plot of the residuals versus the predicted values, separately for the remitted MDD and never-depressed controls.

Regarding childhood trauma, participants were assigned to either one of two groups based on their CTQ scores: a traumatized group having experienced at least one type of clinically significant trauma and a non-traumatized group. Participants were included in the traumatized group if they scored above the following cutoff points, defined by Walker et al. [45], for any of the CTQ subscales: emotional abuse ≥ 10, physical abuse ≥ 8, sexual abuse ≥ 8, emotional neglect ≥ 15, and physical neglect ≥ 8. This categorical trauma variable was entered into the model as fixed factor, together with intervention (placebo, typhoid vaccine without TSST, placebo with TSST or typhoid vaccine with TSST) and time, as well as their interactions. The baseline value of the outcome measure was included as a covariate, as well as study site, body mass index, age, race, use of oral contraceptives, baseline mood score, and use of antidepressants.

The initial fixed effect model was simplified using a stepwise backward elimination. To account for the non-independence between the observation from the same subject, a random effect for subject was included. To account for the five outcome measures investigated in this study, the significance level for the fixed effects was adjusted to 0.01 according to the Bonferroni correction. In case of significant fixed effects of categorical variables or their interaction terms, a post-hoc analysis of least squares means estimates with Tukey’s honestly significant difference (HSD) correction for multiple comparisons was carried out, and these effects are reported within the 95% confidence interval.

## 3. Results

### 3.1. Demographics

As described previously [37], the remitted MDD and control groups differed significantly in their use of antidepressants and their clinical severity scores (MADRS and CTQ) but not in their demographics and baseline biological outcomes (Table 1).

### 3.2. Biological Outcomes

Three individuals dropped out of the study after the first visit. The data of their first visit was included in the analyses. The number of missing data was very limited for cortisol (0.59%), IFN-γ (0.57%), and TNF-α (0.57%). However, this was much more frequent for ACTH (18.81%) and IL-6 (13.14%).

A summary of the general effects of each intervention on the biological outcome measures for each diagnostic group can be found in the previous paper by our research group [37].

#### 3.2.1. Neuroendocrine Outcomes

No main effects of trauma were found on ACTH or cortisol levels in the remitted MDD group or in the never-depressed control participants. For cortisol, however, in the remitted MDD group, a significant interactional effect was found between trauma and intervention (*F* (276.8, 3) = 5.2758; *p* = 0.0015), with significantly higher cortisol values after the combined intervention than after the placebo (Tukey’s *t* = −3.12; *p* = 0.0417; 99% confidence interval (CI) (−0.4689 to 0.0315)) in traumatized individuals and higher cortisol levels after the TSST (Tukey’s *t* = −3.07; *p* = 0.0484; 99% CI (−0.4454 to 0.0336)) than after the placebo in the non-traumatized only (Figure 2). This interactional effect of trauma and intervention was not present in the never-depressed control group.

#### 3.2.2. Inflammatory Outcomes

No effects of trauma were found on IL-6 and IFN-γ levels, neither in the remitted MDD, nor in the never-depressed control group, after Bonferroni correction. For TNF-α, an outlier was detected with Cook’s distance influence of 0.51, jackknife distance of 18.04, and Mahalanobis distance of 12.15. Upon closer inspection, this data point was the measurement at 360 min post-intervention in one individual belonging to the traumatized remitted MDD group. It was very different from the other eight measurements in this individual, therefore, it was excluded from the analyses for the TNF-α outcome. In the remitted MDD group, an interaction effect of trauma with intervention was found (*F* (209.8, 3) = 6.1867; *p* = 0.0005) on TNF-α levels that was not present in the never-depressed control group. In the remitted MDD group, Tukey’s test revealed significantly lower levels of TNF-α after the TSST than after placebo (Tukey’s *t* = 3.46; *p* = 0.0150; 99% CI (0.0088 to 0.1447)) or vaccination (Tukey’s *t* = 3.07; *p* = 0.0484; 99% CI (0.0004 to 0.2095)) within the non-traumatized group and lower levels after the vaccination in the traumatized as compared to the non-traumatized group (Tukey’s *t* = 3.25; *p* = 0.0286; 99% CI (0.0068 to 0.2237)) (Figure 3).

## 4. Discussion

This study sought to assess the effect of childhood trauma on the stress and immune responses to the psychosocial and immunological stressor in remitted MDD. We hypothesized that childhood trauma would lead to differential cortisol and ACTH responses to psychosocial stress in remitted MDD patients. However, contrary to our hypothesis, trauma in itself did not have an effect on any measure of HPA axis function in remitted MDD. Instead, a history of childhood trauma differentiated the cortisol response to the psychosocial and immunological stressor. Though a significant rise in cortisol after a psychosocial stressor was apparent in the non-traumatized group, this was only seen after the addition of immunological to psychosocial stress in traumatized individuals. Thus, it seems that in these individuals, the cortisol response is not stimulated by psychosocial stress alone.

One possible explanation for this unexpected finding is the remitted depressive state of our study group. In the previously cited study by Heim and colleagues [46], reporting HPA axis hyperreactivity to psychosocial stress in abused subjects, this effect was most robust in the subgroup with a current diagnosis of MDD. However, studies in non-depressed maltreated samples have demonstrated a reduced cortisol response to the TSST [47,48]. The fact that this blunted cortisol response to psychosocial stress was seen only in our traumatized remitted (recurrent) MDD group, might be due to an additive effect of trauma and disease state, as chronic and recurrent MDD is associated with a blunted cortisol response [49]. This provides some support for our hypothesis of a trauma-associated vulnerability factor in remitted MDD. However, given the fact that there were only three individuals in our traumatized healthy control group, these results need to be interpreted with the necessary caution. On the other hand, adding an immunological stressor to the TSST does trigger the HPA axis in traumatized individuals with recurrent and remitted MDD. HPA axis activation by inflammatory cytokine release has been described [50]. Thus, the effect of inflammatory stress apparently overrules the reduced cortisol reactivity to psychosocial stress found in traumatized individuals with remitted MDD.

Gender might also have been a confounder in our study. Indeed, studies investigating the cortisol response to psychosocial stress in remitted MDD have reported very different results, from hypo- [51] to hyperresponsiveness [12] as well as a normal cortisol secretion [14]. When taking gender into account, hypocortisolism after psychosocial stress is reported only in female participants [17,52], whereas male participants had a normal [52] to increased cortisol response [17]. Indeed, like in our exclusively female sample, the study reporting cortisol hyporesponsiveness had a majority of female participants (female to male ratio F/M 39:17) [51], whereas the hyperresponsiveness was reported in a primarily male sample (F/M 22:43) [12]. In the study reporting no differences in HPA axis responsiveness, there was an even gender distribution (F/M 37:33) [14].

Regarding the effect of trauma on the inflammatory response to stressors, we expected a proinflammatory state after both stressors in remitted MDD with a history of childhood trauma. Although there was no main effect of childhood trauma, an interaction effect of trauma with intervention was found on TNF-α in the remitted MDD group. Firstly, lower levels of this proinflammatory cytokine after the TSST were observed in the non-traumatized group, but not in traumatized individuals. This is in line with our previous finding of reduced HPA axis responsivity to psychosocial stress and, hence, less immunosuppressive cortisol action in the traumatized group. Secondly, contrary to our hypothesis, trauma was associated with significantly lower levels of TNF-α in response to the vaccination. This is in accordance with the findings by Sorrels et al. [53], who reported that chronic stress can induce a reduced cellular immune response to inflammatory stressors. As such, the TNF-α response to vaccination would be weakened in traumatized individuals.

Several limitations need to be taken into account when considering these results. Firstly, the sample size was relatively small (*n* = 39), and at least for the ACTH, and IL-6, the number of missing data was high. Although the crossover repeated measures design of our study as well as the use of linear mixed models maximized the statistical power, the number of subgroups and dependent and independent variables analyzed was notable. Therefore, smaller effects might not have been detected in this study. On the other hand, the small sample size might have led to an increased false discovery rate. Thus, our study’s results require confirmation in future studies with larger study samples. Also, in our healthy control group, only three individuals reported a history of childhood trauma and, thus, the study’s power might not have been sufficient to detect effects of trauma in our healthy control group. Therefore, any conclusions regarding the specificity of our findings to the MDD group need to be interpreted with caution.

Secondly, this study investigated an all-female sample. Therefore, no conclusions can be drawn from this study regarding the effects of trauma on neuroendocrine and immunological responses to stress in males. The levels of proinflammatory cytokines [54,55] and HPA axis responsiveness to psychosocial stress [56,57] also vary with the different phases of the menstrual cycle. Future studies on the immunological and neuroendocrine effects of childhood trauma in remitted MDD should take this possible confounder into account.

Thirdly, in this study of physiological stress responses to both psychological and immunological stressors and how this may impact on immunological outcomes, only the HPA axis was considered and not the sympathetic nervous system (SNS) or catecholamines. Both stress systems (HPA axis and SNS) are activated in response to psychological stress and activate an immunological response [34]. Moreover, these systems interact during episodes of stress. However, only cortisol is differentially responsive to negative affective states or distress (as induced by the TSST, but also by an episode of major depressive disorder or a history of childhood trauma) [58]. For this reason, this study focused on the HPA axis. Still, future studies should address the relevant question of whether the SNS influences the immunological response to a psychosocial and/or immunological stressor as well as the HPA axis’ response to these stressors and how this may impact on immunological outcomes.

Lastly, this study did not look into (epi)genetic mechanisms involved in the regulation of neuroendocrine and immunological responses in individuals with a history of childhood trauma. However, the epigenetic changes following such history may influence the immune [59] and neuroendocrine responses [60]. Moreover, evidence of (epi)genetic mechanisms involved in the vulnerability (or resilience) of these individuals to subsequent psychiatric (or physical) illness is accumulating [61]. Future studies should address these issues.

## 5. Conclusions

Although there was no main effect of childhood trauma on the neuroendocrine and inflammatory response, there was an interaction of trauma with intervention in remitted MDD patients characterized by significant differences in neuroendocrine and inflammatory responses to psychosocial and immunological stress. More precisely, cortisol responsivity was only seen in traumatized individuals after addition of an immunological stressor to a psychosocial stress induction. A history of childhood trauma was also associated with lower levels of TNF-α after vaccination. These results do provide some support for our hypothesis of a trauma-related vulnerable endophenotype in remitted moderate to severe MDD, associated with differential responses to psychosocial and immunological stress.

## Figures and Tables

**Figure 1 brainsci-09-00375-f001:**
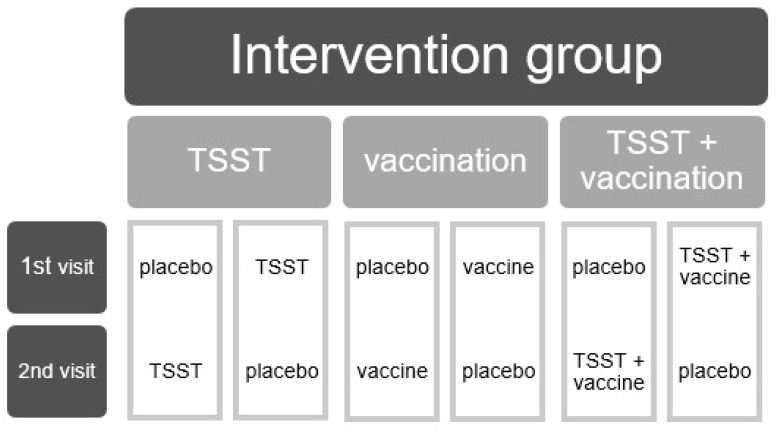
The six treatment conditions of the study’s crossover design. TSST: Trier social stress test.

**Figure 2 brainsci-09-00375-f002:**
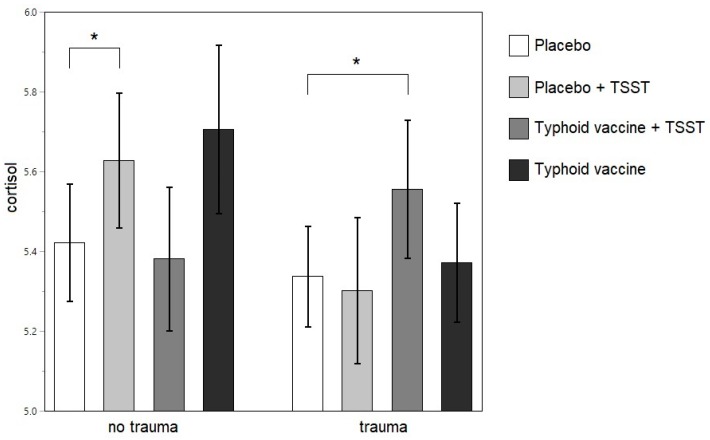
Least square means of the log-transformed cortisol levels after each intervention in the remitted MDD group. * *p* <0.05. TSST: Trier social stress test.

**Figure 3 brainsci-09-00375-f003:**
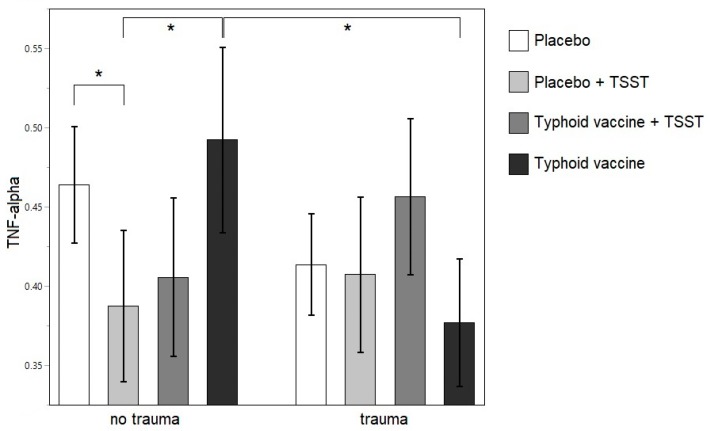
Least square means of the log-transformed TNF-α levels after each intervention in the remitted MDD group. * *p* <0.05. TSST: Trier social stress test.

**Table 1 brainsci-09-00375-t001:** Demographics.

	Remitted MDD Group			Healthy Control Group		
	No Trauma (*n* = 9)	Trauma (*n* = 12)		No Trauma (*n* = 15)	Trauma (*n* = 3)	
	Median (*IQR*)	Median (*IQR*)	*p* Value	Median (*IQR*)	Median (*IQR*)	*p* Value
**Age**	38.0 (11.5)	29.5 (8.5)	0.0639	30.0 (13.0)	32.0 (18.0)	0.9524
**MADRS**	3 (12.0)	5 (9.5)	0.7207	0.0 (1.0)	0.0 (2.0)	1.0000
**Baseline ACTH**	2.162 (1.995)	1.953 (1.595)	0.8940	1.847 (1.847)	2.057 (1.097)	0.8930
**Baseline cortisol**	248.218 (113.717)	269.462 (254.402)	0.8590	277.923 (333.931)	135.835 (151.745)	0.1551
**Baseline IL-6**	0.485 (0.185)	0.510 (0.465)	1.0000	0.474 (0.157)	0.647 (1.403)	0.1116
**Baseline IFN-γ**	4.363 (13.665)	3.780 (4.388)	0.8036	3.590 (4.230)	4.320 (82.897)	0.9057
**Baseline TNF-α**	1.470 (0.706)	1.490 (0.707)	0.6959	1.341 (0.453)	1.780 (1.172)	0.2863
	**Mean (*SD*)**	**Mean (*SD*)**	***p* Value**	**Mean (*SD*)**	**Mean (*SD*)**	***p* Value**
**BMI**	23.9 (3.0)	24.1 (2.8)	0.8694	22.0 (2.9)	24.5 (4.3)	0.4210
**CTQ**	30.6 (3.6)	51.8 (16.8)	**0.0011**	29.4 (3.1)	41.7 (10.7)	0.1834
	**Column Percentage**	**Column Percentage**	***p* Value**	**Column Percentage**	**Column Percentage**	***p* Value**
**Oral contraceptives**	66.67	25.00	0.0562	66.67	66.67	1.000
**Ethnicity**			0.2367			0.1797
African	11.11	0.00		0.00	0.00	
Maghrebi	0.00	0.00		6.67	33.33	
European	88.89	100.00		93.33	66.67	
**Antidepressants**	55.56	75.00	0.3496	00.00	00.00	-

Abbreviations: MDD: major depressive disorder; IQR: interquartile range; MADRS: Montgomery–Åsberg depression rating scale; ACTH: adrenocorticotropic hormone; IL-6: interleukin 6; IFN-γ: interferon gamma; TNF-α: tumor necrosis factor alpha; SD: standard deviation; BMI: body mass index; CTQ: Childhood Trauma Questionnaire.

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
