# Peer review of "Neuroendocrine and Inflammatory Effects of Childhood Trauma Following Psychosocial and Inflammatory Stress in Women with Remitted Major Depressive Disorder"

_brainsci, 2019, doi:10.3390/brainsci9120375_

Round 1
Reviewer 1 Report
This is a very interesting study. The authors hypothesized that childhood trauma would be associated with differential cortisol and ACTH responses to a psychosocial stressor in remitted MDD patients. Additionally, they hypothesized that given the pro-inflammatory state associated with childhood trauma and the fact that immune cells can become resistant to the anti-inflammatory effects of glucocorticoids under acute psychosocial stress, levels of inflammatory cytokines would be elevated after both stressors. The study design is robust, the utilized methods state of the art and the statistical analyses are appropriate.
Some issues should be addressed in a revised manuscript:
Please discuss possible epigenetic changes following child hood trauma and the role of genetic vulnerability/resilience as additional factors that can influence the expression of the biomarkers you tested. Please address the possible confounding effects of female hormones on the measured biomarkers since you did not specify whether the specific phase in the menstrual cycle was controlled for. Please comment on the possible confounding effect of the contraceptive method used, assuming it was hormone-based. Please comment, if you documented, the number of previous episodes your MDD cohort had suffered and whether a higher number of episodes might have exerted an influence on the biomarkers you assessed. Please comment whether the MADRS ratings were conducted by rates blind to the study design. Were the injected substances given intravenously or subcutaneously? In section 2.2 the sentence starting with “The intervention sequences existed of…” should read “The intervention sequences consisted of…”
Author Response
We would like to thank the reviewer for the interesting comments. We have formulated a reply to each comment below, stating clearly when and where changes were made to the manuscript. Included you will also find a revised version of the manuscript, with all changes marked by track changes.
Comment 1: Please discuss possible epigenetic changes following child hood trauma and the role of genetic vulnerability/resilience as additional factors that can influence the expression of the biomarkers you tested.
Answer 1: There is indeed evidence for epigenetic changes that mediate the relationship between childhood trauma and cortisol reactivity to stress [1], and inflammation as well [2]. Furthermore, the interaction between these epigenetic processes, genetic predisposition and the neuroendocrine and immune systems is thought to play a role in the vulnerability (or resilience) of individuals who suffered childhood maltreatment to psychiatric (as well as physical) illness [3]. Though the (epi)genetic mechanisms regulating the HPA axis and immune system were not the scope of this paper, future studies should address these issues.
We added the following limitation to our Discussion section:
“Lastly, this study did not look into (epi)genetic mechanisms involved in the regulation of neuroendocrine and immunological responses in individuals with a history of childhood trauma. However, the epigenetic changes following such history may influence the immune (Cattaneo et al., 2015) and neuroendocrine responses (Houtepen et al., 2016). Moreover, evidence of (epi)genetic mechanisms involved in the vulnerability (or resilience) of these individuals to subsequent psychiatric (or physical) illness is accumulating (Ehlert, 2013). Future studies should address these issues.”
Comment 2: Please address the possible confounding effects of female hormones on the measured biomarkers since you did not specify whether the specific phase in the menstrual cycle was controlled for.
Answer 2: Levels of inflammatory cytokines indeed vary across the different phases of the menstrual cycle [4,5], as does cortisol responsiveness to psychosocial stress [6,7]. We did not document menstrual cycle phase and therefore we could not control for it. However, it is known that the cortisol response to psychosocial stress is influenced by sex steroid hormones, in fact: cortisol reactivity is smaller in women compared to men in this case [8].
To address this issue, we added the following limitation to our Discussion section:
“The levels of proinflammatory cytokines (Evans and Salamonsen, 2012; Salamonsen and Lathbury, 2000) and HPA axis responsiveness to psychosocial stress (Maki et al., 2015; Montero-López et al., 2018) also vary with the different phases of the menstrual cycle. Future studies on the immunological and neuroendocrine effects of childhood trauma in remitted MDD should take this possible confounder into account.”
Comment 3: Please comment on the possible confounding effect of the contraceptive method used, assuming it was hormone-based.
Answer 3: The levels of proinflammatory cytokines [9] and cortisol response to psychosocial stress [10] are indeed influenced by oral contraceptives. Some of our participants were using oral (hormone-based) contraceptives. As illustrated by Table 1, the differences in rates of oral contraceptive use between trauma groups were not significant. Also, we added oral contraceptive use as a covariate in our linear mixed models, but could not find a significant effect for any of the biological outcome measures. Therefore it was removed from each model by stepwise backward elimination.
Comment 4: Please comment, if you documented, the number of previous episodes your MDD cohort had suffered and whether a higher number of episodes might have exerted an influence on the biomarkers you assessed.
Answer 4: The precise number of previous MDD episodes of our MDD cohort was not documented. Given the association between lasting HPA axis changes and chronic or recurring MDD [11].and the documented bidirectional link between MDD and inflammation [12,13], we cannot rule out the possibility that the number of previous depressive episodes may influence the levels of the biomarkers assessed in this study. However, given that only participants with remitted recurrent MDD were recruited, all experienced at least two episodes of MDD. Also, we excluded participants taking more than one antidepressant. Based on these inclusion criteria, we expect our study population to be fairly homogenous concerning the chronicity and number of previous MDD episodes.
Comment 5: Please comment whether the MADRS ratings were conducted by rates blind to the study design.
Answer 5: The MADRS rating was completed on the first study visit by the investigators for eligibility screening (scores had to be below 15), as specified in our Materials and Methods section, paragraph 2.1. Subjects). Only the participants were blinded to the study design.
Comment 6: Were the injected substances given intravenously or subcutaneously?
Answer 6: Both the typhoid vaccine and the placebo (NaCl 0.9%) were injected intramuscularly. This was further specified in the text under paragraph 2.2. Study Design:
“The immunological stressor consisted of an intramuscular injection with the typhoid vaccine (0.5 mL containing 25 μg S. typhi capsular polysaccharide; Typhim ® Vi; Sanofi Pasteur MSD, Diegem, Belgium). The placebo condition consisted of an intramuscular injection with 0.5 mL NaCl 0.9%.”
Comment 7: In section 2.2 the sentence starting with “The intervention sequences existed of…” should read “The intervention sequences consisted of…”
Answer 7: This was adjusted in the text as requested.
References:
Houtepen, L.C.; Vinkers, C.H.; Carrillo-Roa, T.; Hiemstra, M.; van Lier, P.A.; Meeus, W.; Branje, S.; Heim, C.M.; Nemeroff, C.B.; Mill, J.; et al. Genome-wide DNA methylation levels and altered cortisol stress reactivity following childhood trauma in humans. Nat. Commun. 2016, 7, 10967. Cattaneo, A.; Macchi, F.; Plazzotta, G.; Veronica, B.; Bocchio-Chiavetto, L.; Riva, M.A.; Pariante, C.M. Inflammation and neuronal plasticity: a link between childhood trauma and depression pathogenesis. Front. Cell. Neurosci. 2015, 9, 40. Ehlert, U. Enduring psychobiological effects of childhood adversity. Psychoneuroendocrinology 2013, 38, 1850–1857. Evans, J.; Salamonsen, L.A. Inflammation, leukocytes and menstruation. Rev. Endocr. Metab. Disord. 2012, 13, 277–288. Salamonsen, L.A.; Lathbury, L.J. Endometrial leukocytes and menstruation. Hum. Reprod. Update 2000, 6, 16–27. Maki, P.M.; Mordecai, K.L.; Rubin, L.H.; Sundermann, E.; Savarese, A.; Eatough, E.; Drogos, L. Menstrual cycle effects on cortisol responsivity and emotional retrieval following a psychosocial stressor. Horm. Behav. 2015, 74, 201–208. Montero-López, E.; Santos-Ruiz, A.; García-Ríos, M.C.; Rodríguez-Blázquez, M.; Rogers, H.L.; Peralta-Ramírez, M.I. The relationship between the menstrual cycle and cortisol secretion: Daily and stress-invoked cortisol patterns. Int. J. Psychophysiol. 2018, 131, 67–72. Lovallo, W.R.; Buchanan, T.W. Stress Hormones in Psychophysiological Research: Emotional, Behavioral, and Cognitive Implications. In Handbook of Psychophysiology; Cacioppo, J.T., Tassinary, L.G., Berntson, G.G., Eds.; Cambridge University Press: Cambridge, 2017; pp. 465–494. Sikora, J.; Mielczarek-Palacz, A.; Kondera-Anasz, Z.; Strzelczyk, J. Peripheral blood proinflammatory response in women during menstrual cycle and endometriosis. Cytokine 2015, 76, 117–122. Mordecai, K.L.; Rubin, L.H.; Eatough, E.; Sundermann, E.; Drogos, L.; Savarese, A.; Maki, P.M. Cortisol reactivity and emotional memory after psychosocial stress in oral contraceptive users. J. Neurosci. Res. 2017, 95, 126–135. Juruena, M.F. Early-life stress and HPA axis trigger recurrent adulthood depression. Epilepsy Behav. 2014, 38, 148–159. Kiecolt-Glaser, J.K.; Derry, H.M.; Fagundes, C.P. Inflammation: depression fans the flames and feasts on the heat. Am. J. Psychiatry 2015, 172, 1075–91. Irwin, M.R.; Slavich, G.M. Psychoneuroimmunology. In Handbook of Psychophysiology; Cacioppo, J.T., Tassinary, L.G., Berntson, G.G., Eds.; Cambridge University Press: Cambridge, 2017; pp. 377–397.
Reviewer 2 Report
The study design is interesting and tried to solve difficult Problems, adding to previous studies because of the randomized crossover design and type of approaches.
The solution however is open and the interpretation difficult , also contrary to expected by the aithiors (as expressed by themselves) , which could point to different interpretations and insights . Insofar the results are even important to report, ie. the main results as reported.
I have several questions first:
for me it was not ablolute clear whether also the controls underwent the Intervention, but I felt that not, in figure 2 is no clear presentation of the Control Group, thesame in figure 3.
Thus I conclude Intervention was done only in MDD patients. Is that true?In any case should be made very clear in text and figures.
Given no Intervention was done in controls, it is the morte relevant to consider the absolute values in controls, which are divergent for MDD versus controls with respect to the aspect non-trauma versus Trauma for baseline ACTH and Cortisol, at least in the same direction and for cortisol considerable difference (not significant indicated , but unclear how calculated) and because both parameters diverge in the same direction maybe these should be evaluated combined!!
When these Points are more clear I maybe able to comment on the Interpretation.
Author Response
We would like to thank the reviewer for the interesting comments. We have formulated a reply to each comment below, stating clearly when and where changes were made to the manuscript. Included you will also find a revised version of the manuscript, with all changes marked by track changes.
Comment 1: for me it was not ablolute clear whether also the controls underwent the Intervention, but I felt that not, in figure 2 is no clear presentation of the Control Group, thesame in figure 3. Thus I conclude Intervention was done only in MDD patients. Is that true?In any case should be made very clear in text and figures.
Answer 1: All participants, both (remitted) MDD patients and healthy controls underwent the intervention (according to the crossover design explained in Figure 1). We added the following specification to paragraph 2.2. Study Design:
“Subjects considered eligible for participation, both healthy control participants and participants with remitted MDD, were computer-randomized to one of six possible intervention sequences administered over the course of two study visits, separated by a seven to fourteen days washout period.”
Because an interaction effect of trauma was found only in the remitted MDD group, we show only figures of the remitted MDD group for illustrational purposes. We agree that this may give cause to some confusion, therefore we specify in the captions accompanying figures 2 and 3 that they represent the results of the remitted MDD group:
“…after each intervention in the remitted MDD group.”
Also, we added a specification in the text under paragraph 3.2.1. Neuroendocrine Outcomes:
“This interactional effect of trauma and intervention was not present in the never-depressed control group.”
We also added the following specification under paragraph 3.2.2. Inflammatory Outcomes:
“In the remitted MDD group Tukey’s test revealed significantly lower levels of TNF-α after the TSST than after placebo (Tukey’s t = 3.46; p = 0.0150; 99% CI [0.0088 to 0.1447]) or vaccination (Tukey’s t = 3.07; p = 0.0484; 99% CI [0.0004 to 0.2095]) within the non-traumatized group and lower levels after the vaccination in the traumatized as compared to the non-traumatized group (Tukey’s t = 3.25; p = 0.0286; 99% CI [0.0068 to 0.2237]) (Figure 3).”
Comment 2: Given no Intervention was done in controls, it is the morte relevant to consider the absolute values in controls, which are divergent for MDD versus controls with respect to the aspect non-trauma versus Trauma for baseline ACTH and Cortisol, at least in the same direction and for cortisol considerable difference (not significant indicated , but unclear how calculated) and because both parameters diverge in the same direction maybe these should be evaluated combined!!
Answer 2: For the results of the comparison between the baseline biological outcomes in the remitted MDD group and the healthy control group we refer to a previous paper by our research group and added the following specification in paragraph 3.1. Demographics:
“As described previously [1], the remitted MDD and control groups differed significantly in their use of antidepressants and their clinical severity scores (MADRS and CTQ) but not in their demographics and baseline biological outcomes.”
In the current study, we used non-parametric tests to investigate differences in baseline biological outcomes between trauma groups within the remitted MDD and the healthy control group (as illustrated by table 1). These differences were not statistically significant: though the medians may appear to diverge, the high interquartile ranges cause the difference to be non-significant. This larger error on the baseline outcome measures is probably due to the small number of total participants and the fact that the baseline value of each outcome was measured only once on each study visit (thus leading to a limited number of baseline data). This is less of a problem for our main analyses, however, because the repeated measurements of each biological outcome following an intervention, as well as the use of linear mixed models maximized the statistical power.
References:
Niemegeers, P.; De Boer, P.; Dumont, G.J.H.; Van Den Eede, F.; Fransen, E.; Claes, S.J.; Morrens, M.; Sabbe, B.G.C. Differential Effects of Inflammatory and Psychosocial Stress on Mood, Hypothalamic-Pituitary-Adrenal Axis, and Inflammation in Remitted Depression. Neuropsychobiology 2016, 74, 150–158.
Round 2
Reviewer 2 Report
is ok now